# Exploring Compositionality in Vision Transformers using Wavelet Representations

## Abstract

Insights into the workings of the transformer model have been elicited by analysing its representations when trained and tested on language data. In this paper, we turn an analytical lens on the representations learnt by the Vision Transformer (ViT) encoder. Specifically, we present a framework to test for compositionality in the ViT encoder. This framework is analogous to the compositionality setting proposed for representation learning in Andreas (2019). Crucial to drawing this analogy is the Discrete Wavelet Transform (DWT). The DWT is a simple yet effective tool for establishing the notion of input-dependent primitives in the vision setting. Our analysis explores the compositional structure induced by the DWT. Several tests are conducted to quantify the extent to which the encoder representations respect the compositional structure of the input space. This empirical analysis reveals interesting insights into compositionality in ViTs. One such insight is that the primitives from a one-level DWT representation of images satisfy compositionality in the representation space.

## 1 Introduction

Vision Transformers (ViTs), in their supervised (Dosovitskiy et al., 2021), self-supervised (Caron et al., 2021) and unsupervised (He et al., 2022) variants, have spurred the development of computer vision applications that deliver consistently good performance. ViTs leverage the power of transformers, originally popularized in natural language processing tasks, to directly process images without relying on convolutions. This fusion of the transformer architecture with computer vision has opened new vistas for understanding and processing visual data. Image classification (Dosovitskiy et al., 2021), object detection (Li et al., 2022), semantic segmentation (Strudel et al., 2021), and image captioning and generation (Radford et al., 2021) are a few examples of computer vision tasks where ViTs have delivered state-of-the-art performance.

It is natural to wonder why ViTs deliver the performance they do despite their origins in language models. Given their prevalence as backbones for generating image embeddings for various downstream tasks, we focus our investigation on the representations themselves. Several works have investigated the inner workings of the ViT. (Raghu et al., 2021) show that the representations of ViT encoder layers are much more uniform than the CNN-based architectures. (Park & Kim, 2022) sheds light on the Multi-head Self Attention block and its optimization. (Bhojanapalli et al., 2021) test the ViT's robustness to input and model perturbations. Their correlation analysis led to interesting findings about ViT models organizing themselves into correlated groups. Our motivation is along the lines of such studies attempting to understand the representations learned by ViTs and make them more explainable. Questions stemming from the basis of these studies led us to some interesting insights. The main contributions of this paper are summarized as follows.

1. A general framework for testing compositionality in ViT encoder representations that is analogous to the framework proposed by Andreas (2019) for representation learning.

2. The use of the Discrete Wavelet Transform (DWT) to generate basis sets (input-specific primitives) for images. To the best of our knowledge, previous works have not used this approach to analyse ViTs.

3. Promising empirical results that demonstrate compositionality in the encoder representations of the ViT. Interestingly, our analysis reveals that ViT patch representations at the last

encoder layer are compositional with respect to the DWT primitives induced by a one-level decomposition.

## 2 BACKGROUND

### 2.1 VISION TRANSFORMERS

Inspired by the success of transformers Vaswani et al. (2017) in natural language processing, (Dosovitskiy et al., 2021) successfully transferred its capabilities to vision tasks. The input image is divided into patches, and each patch is tokenized. Positional embeddings are added to preserve the spatial location of the patch. ViTs append a special CLS token to the input embeddings which is used for image classification. The dimension of all the patch representations stay constant throughout the encoder layers which gives the ViT model a lot of flexibility.

### 2.2 COMPOSITIONALITY IN REPRESENTATION LEARNING

Representational compositionality has been a field of study since the days of the connectionist approach (Fodor & Pylyshyn, 1988; Chalmers, 1990). Its linguistic origins still make themselves known in current research, with most investigations focusing on representations in natural language processing tasks and models (Chen et al., 2023; Li et al., 2023; Dziri et al., 2023). (Janssen, 2001) defines the principle of compositionality as "the meaning of a compound expression is a function of the meaning of its parts and of the syntactic rule by which they are combined". The notions of *meaning* and *syntactic rules* naturally lend themselves to the study of compositionality for language model representations.

Formally, if we abstract away the details of the input as well as those of downstream task, a compositional representation learner is one that learns a homomorphism between the space of its inputs and the space of its representations (Andreas, 2019), where a homomorphism $\phi : H \to G$ is an injective map between two groups $(H, \cdot)$ and $(G, \oplus)$, such that if $\phi(h_1) = g_1$ and $\phi(h_2) = g_2$ for $h_1, h_2 \in H$ and $g_1, g_2 \in G$, then $\phi(h_1 \cdot h_2) = g_1 \oplus g_2$.

The study of compositionality for language inputs is divided into two broad classes (Hupkes et al., 2020), those that study the capacity for neural networks to generalize compositionally on artificial data and those that study the compositional nature of models trained on natural data. Investigations into the compositional nature of pretrained models are motivated, at least partly, by interpretability. A model that can break apart its input into meaningful pieces and reconstruct it in a human-understandable manner is more interpretable than one that does not do so. It is with interpretability in mind that we pursue our investigations into the representations learned by ViT.

Investigations into the compositional nature of transformer models in the NLP domain usually decompose the input space into a dictionary of words. This dictionary acts as the fundamental set using which all sentences are created. However, in the image domain, it is difficult to construct a dictionary of *visually meaningful* images since the image space is continuous. In other words, we cannot construct a dictionary with infinite cardinality. This difficulty is additionally compounded by the uninterpretable nature of the canonical basis in image space, the set of $H \times W \times C$ matrices with every element being 0 except for a single 1 at some position. Thus, we propose a different approach to decompose an image into its visually meaningful primitives, turning to analytical tools from signal processing.

### 2.3 DISCRETE WAVELET TRANSFORM (DWT)

While the Fourier series and Fourier transform are excellent tools for analyzing the frequency spectrum of images, they do not provide localization in the pixel domain. In other words, the Fourier spectrum of an image is not visually meaningful. The DWT Daubechies (1992) stands out among time-frequency analysis tools due to its unique ability of time-frequency localization. Specifically, the sub-band decomposition of an image obtained by applying the DWT forms an ideal input-dependent primitive. The invertibility of the sub-band decomposition enables lossless reconstruction making the DWT our tool of choice for compositionality analysis. After the introduction of ViTs, the DWT has been used for lossless downsampling to address their efficiency-vs-accuracy tradeoff (Yao et al., 2022).

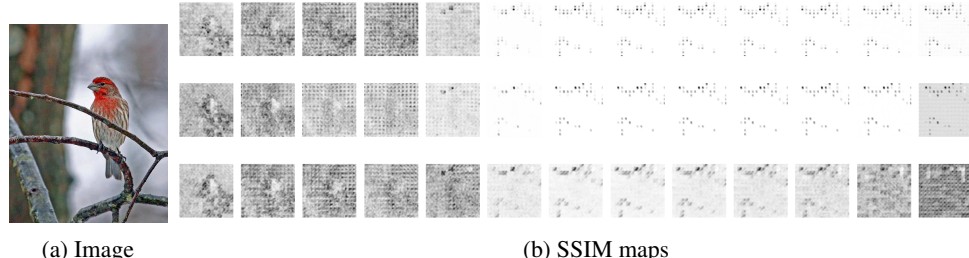

(a) Image        (b) SSIM maps

Figure 1: SSIM maps for each channel (R,G,B). For each encoder layer output, the original image's representation is compared with the composed image representation. The SSIM maps shown here are **after** comparison. There is no immediate notion of compositionality present visually.

Zhang et al. (2024) also employs the DWT to improve the quality of the input in a transformer-based network. However, to our knowledge, the DWT has not been used to explore compositionality in ViTs.

### 2.4 COMPOSITIONALITY OF IMAGE REPRESENTATIONS

When inputs belong to the pixel space, and a neural network learns input representations, the groups across which compositionality is studied are vector spaces. These spaces need to be equipped with a binary operation that satisfies the group axioms, the natural choice being vector addition.

A homomorphism between two vector spaces $V$ and $W$ reduces to a linear map $T : V \rightarrow W$. This map is completely defined by its action on the basis set $\{v_1, v_2, ...\}^1$ of $V$. Thus, we come to our central assertion that the ideal compositional representation learner is one that is capable of preserving the structure of vector addition between the pixel and representation spaces. This behaviour is obviously not preserved in real models trained on real data, not the least because of the deep nesting of non-linearities. Thus, the question becomes one of *finding* a composition method in latent space instead of asserting that it is simply addition.

To quantify this behaviour, we aim to study the evolution of the representations of the basis set as it moves through the model. In latent space, we recompose the primitives' representations in a manner analogous to pixel space and compare the resultant representations with that of the original image. This method acts as a lens into the manifolds learned by each encoder layer. To make this analysis manageable, we focus on compositionality in the last encoder layer.

## 3 COMPOSITIONALITY ANALYSIS

### 3.1 CAPTURING COMPOSITIONALITY

The wavelet reconstruction in the image space gives back the original image without any loss of information. The obvious approach to check the compositionality of the model would be to see how the reconstruction behaves in the representation space of the encoder layers. Suppose $d_a, d_b, d_c, d_d$ are Level 1 wavelet coefficients of image $I$ such that

$$DWT(I) = (d_a, (d_b, d_c, d_d))$$

$d_a, d_b, d_c, d_d$ can also be referred as Low-Low (LL), Low-High(LH), High-Low(HL) and High-High(HH) frequency bands. Then,

$$D_a = IDWT(d_a); D_b = IDWT(d_b); D_c = IDWT(d_c); D_d = IDWT(d_d)$$

$$I = D_a + D_b + D_c + D_d \tag{1}$$

where IDWT is the Inverse Discrete Wavelet Transform and $D_a, D_b, D_c, D_d$ are the corresponding reconstructed images of individual coefficients. We analyse if such composition (1) of the reconstructed encoder layer representations also gives the image's encoder layer representation. We identify

---

[1]We also refer to bases as *primitives*

two metrics, Structural Similarity Index metric (SSIM) (Wang et al., 2004) and Centered Kernel Alignment(CKA) (Kornblith et al., 2019) to measure the similarities between the image's encoder layer representation and the composition of the reconstructed layer encoder representation. SSIM is a perceptual metric and takes into account local patterns of pixel intensities, their correlation, and spatial arrangements. CKA is used to compare the similarity between two sets of high-dimensional feature vectors (often from neural network layers). Using these metrics we perform two analyses,

1. We use the SSIM map (Wang et al., 2004) to visualize any structural similarities between the original and the composed representations. To do this, we reshape the encoder layer representation $E_L(I)^{N-1 \times D} \longrightarrow E_L(I)^{W \times H \times C}$ where $N$ is the number of tokens ($N-1$ to exclude the `CLS` token), and $D$ is the hidden dimension of encoder layer. We measure the SSIM across the channels.

2. We plot the CKA (Kornblith et al., 2019) scores for all encoder layers between the composed representation and the image representation. To do this experiment, we take a sample of 10k images(10 images per class) from the imagenet-1k dataset and average the CKA scores over all encoder layers.

Figure 1 presents the SSIM maps computed for a sample image, and Figure 2 presents the CKA scores averaged over 10K images from the ILSVRC validation set over the ViT-Base representations. Both the maps and the scores, unsurprisingly, do not provide any evidence of compositionality or structural similarity between the representations. It is difficult to digest that by simply adding the individual wavelet representations would perfectly give back the original image's representation, but this also invites the possibility that the reconstruction of these primitives in the representation space is different from the reconstruction in the image space. This leads to see if we can learn such a composition function, if it exists, by relaxing the constraint that each wavelet representation has to be equally weighted.

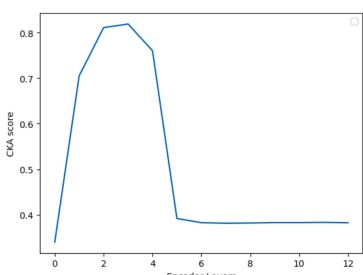

Figure 2: CKA scores of original vs composed representations at various encoder layers of ViT-B averaged over 10K images.

### 3.2 DRAWING PARALLELS FROM EXISTING WORKS

The inspiration for a framework to study compositionality in ViTs stems from the work by Andreas (2019). The paper offers a framework to measure compositionality in deep learning models, particularly neural networks. In the context of this paper, compositionality refers to the ability of system to represent complex ideas using simpler concepts. The framework evaluates a metric to measure how well an explicitly compositional model $\hat{f}_\eta$ can approximate a complex model $f$. In order to draw parallels, we summarize our understanding of their framework, with corresponding analogies to our approach:

**1) Representations**: We consider a *model* $f : \mathcal{X} \longrightarrow \Theta$, where $\mathcal{X}$ is a dataset of observations and $\Theta$ is a space of representations $\theta$. The representations produced by $f$ are analyzed via the proposed framework for compositional behavior.
**Analogy**: *model* $f : \mathcal{X} \longrightarrow \Theta$ is the ViT model, $\mathcal{X}$ is the dataset consisting of images and $\Theta$ is the ViT encoder representation space with representations $\theta$.

**2) Derivations**: The inputs from dataset $\mathcal{X}$ can be realised with tree structured derivations $d$ defined by a finite set $\mathcal{D}_0$, consisting of primitives, along with a bracketing operation $\langle ., . \rangle$ such that if $d_i$ and $d_j$ are derivations, $\langle d_i, d_j \rangle$ is also a derivation. A derivation oracle $D : \mathcal{X} \longrightarrow \mathcal{D}$ is used to extract the derivatives.

**Analogy**: The DWT offers a way to construct the inputs from a tree structure (Figure 3) of its respective wavelet coefficients. Although the set of all such wavelet primitives is infinite (2.2), if $d_i$ and $d_j$ are derivations (read, wavelet primitives), then their combination is also a derivation. Our oracle $D$ is the DWT itself which constructs the derivation of an image.

**3) Compositionality**: The model $f$ is compositional if it is a homomorphism from input space to representation space. A composition operation $*: \theta_a * \theta_b \mapsto \theta$ is defined such that for any $x$ with $D(x) = \langle D(x_a), D(x_b) \rangle$,

$$f(x) = f(x_a) * f(x_b)$$

Although such primitives whose composition exactly reproduces the model's prediction may not exist, can the candidate primitives approximate the input's representation? If a learnable compositional model $\hat{f}_\eta$ with parameters $\eta$ can approximate $f$ it could serve as a measure of compositionality for the model.

**Analogy**: The model $f : \mathcal{X} \longrightarrow \Theta$ is the ViT model. We can consider the ViT model till an intermediate encoder layer $l$ such that $f_l : \mathcal{X} \longrightarrow \Theta$ also operates from the same input space into the encoding layer $l$. Then the compositional model $\hat{f}_\eta(d) : \mathcal{X} \longrightarrow \Theta$ can be viewed as an approximation of encoder layer $f_l$ from the input space. With this perspective, we can study the compositionality of any encoder layer of the ViT architecture.

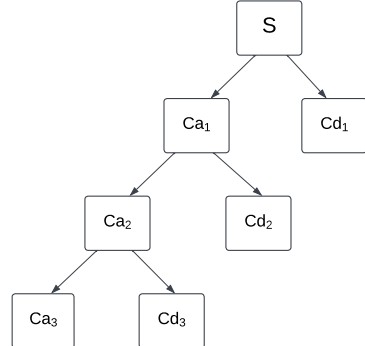

Figure 3: Tree structure of Discrete Wavelet Transform. S represents the input signal. $Ca_i, Cd_i$ represent the approximate and detail coefficients of $i^{th}$ level.

### 3.3 COMPOSITIONALITY FRAMEWORK FOR VITS

To generate the primitives set for pixel space, we turn to the 2D Discrete Wavelet Transform (DWT). The DWT has long been employed as a tool for time-frequency analysis due to its invertibility and for its unique ability to capture temporal resolution. It is a great fit for our work since its exact reconstruction property makes it an ideal tool for studying compositionality. Given any $W \times H$ image $I$, it can be discretely represented by its wavelet coefficients as,

$$I_{W \times H} = \sum_{i=0}^{W-1} \sum_{j=0}^{H-1} A_{M,i,j} \phi_{M,i,j} + \sum_{m=1}^{M} \sum_{i=0,j=0}^{W-1,H-1} \sum_{k=1}^{3} D_{m,i,j}^{k} \psi_{m,i,j}^{k} \qquad (2)$$

where $A_{M,i,j} = \langle I_{W \times H}, \phi_{M,i,j} \rangle$ and $D_{M,i,j}^{k} = \langle I_{W \times H}, \psi_{M,i,j}^{k} \rangle$ are the approximation and detail coefficients respectively, and $k$ is the sub-band index, and $\phi, \psi$ are the approximation and detail wavelet bases respectively. An orthogonal decomposition is assumed in this work. The first term is the approximation of the image at level $M$ and the second term represents all of the detail coefficients from level 1 to $M$, which, when added to the approximation coefficient, gives finer details.

Let $E_l : \mathbb{R}^{W \times H \times C} \longrightarrow \mathbb{R}^{N \times D}$ be a function that accepts an image input of dimension $W \times H \times C$ and outputs a set of $N$ token vectors of length $D$ from the $l^{th}$ layer of a vision transformer with $L$ encoder layers (i.e., $1 \le l \le L$).

We investigate the following question to check for the compositionality of a ViT model. Is

$$\sum_{l=1}^{L} ||E_l(I) - (E_l(\sum_{i=0}^{W-1} \sum_{j=0}^{H-1} A_{M,i,j} \phi_{M,i,j}) + \sum_{m=1}^{M} E_l(\sum_{i=0,j=0}^{W-1,H-1} \sum_{k=1}^{3} D_{m,i,j}^{k} \psi_{m,i,j}^{k}))||_2 = 0? \quad (3)$$

The preliminary analysis presented in Figs. 1 and 2 gives a negative answer to the above question.

Considering the highly non-linear nature of the ViT model and the high dimensional space of the representations, we modify the question to

$$E_l(I) \approx g_\eta(E_l(\sum_{i=0}^{W-1} \sum_{j=0}^{H-1} A_{M,i,j} \phi_{M,i,j}), \sum_{m=1}^{M} E_l(\sum_{i=0,j=0}^{W-1,H-1} \sum_{k=1}^{3} D_{m,i,j}^{k} \psi_{m,i,j}^{k}))? \qquad (4)$$

i.e., can we approximate the composition of the primitive representations to the original representations at layer $l$ of the encoder where $g_\eta(.)$ is a learnable composition function (with parameters $\eta$) of the primitive representations? To emphasize, $g_\eta(.)$ attempts to find the best possible *linear combination* of the primitive representations.

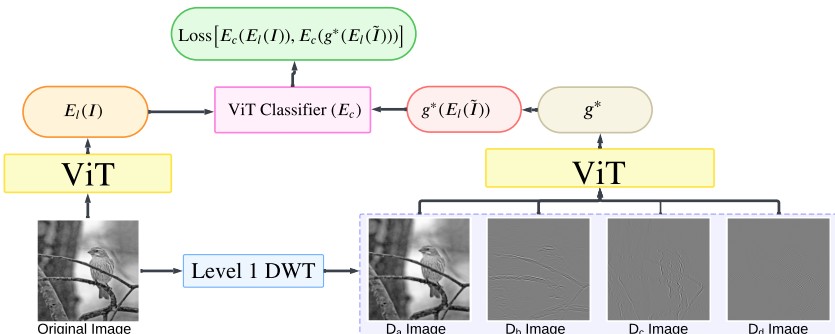

Figure 4: Overview of the proposed compositionally framework for ViTs. The figure presents learning the composition function for Level 1 DWT decomposition. $D_a$, $D_b$, $D_c$, $D_d$ are the coefficients of the wavelet decomposition discussed in 1 .

We argue that popular distance metrics between these two representations might not be a reliable way of measuring similarity due to the curse of dimensionality. Instead, we aim to look at the final layer output of the classification head and minimize the loss between the original image's final output and the approximate *linearly combined* final output. Since we are not modifying any of the ViT model's parameters while training this approximate model, we affirm that all our analyses are post-hoc and still viable probes for understanding the pretrained ViT model. Hence, our reformulated question to check the compositionality becomes,

$$\eta^* = \arg\min_{\eta} \mathcal{L}[E_c(E_l(I)), E_c(g_{\eta}^*(\tilde{I}))], \tag{5}$$

$$\tilde{I} = (E_l(\sum_{i=0}^{W-1}\sum_{j=0}^{H-1} A_{M,i,j}\phi_{M,i,j}), \sum_{m=1}^{M} E_l(\sum_{i=0,j=0}^{W-1,H-1}\sum_{k=1}^{3} D_{m,i,j}^k \psi_{m,i,j}^k)) \tag{6}$$

where $\mathcal{L}$ is the loss function, $E_c : \mathbb{R}^{N \times D} \longrightarrow \mathbb{R}^{1 \times C}$ is the classifier head of the ViT and $E_l$ is the encoder layer output. The composition function $g_{\eta}(.)$ with optimal parameters $\eta^*$ is denoted $g_{\eta}^*(.)$. Figure 4 visualizes our proposed framework to learn the composition function $g_{\eta}^*(.)$. For readability, we drop the subscript and simply use $g^*(.)$.

## 4 EXPERIMENTAL SETUP AND RESULTS

The framework discussed in the previous section was implemented with the following details. We use the ImageNet-1k (Deng et al., 2009) dataset, which consists of 1000 classes. A sample of 50 images from each class is taken, and the entire set (50,000 samples) is divided into a 60:20:20 train:val:test split. To generate the dataset required for the composition function $g^*$, we extract the encoder layer representations $E_l(\tilde{I})$, specifically the cls token of the representation, for each wavelet coefficient. These tokens then act as input to $g^*$, and we get the composed cls token. The target is the final classification layer output of the original image (not the ground truth classification label). We optimize the loss (Cross-Entropy loss) only to learn $g^*$'s parameters while keeping the ViT's parameters frozen. The models are trained for 100 epochs using SGD optimizer with a learning rate of 0.001.

We restrict our analysis to two levels of DWT decomposition using two different wavelet bases (Haar and db4). We also include two variants of ViT, ViT-B, and ViT-L, both pretrained on the ImageNet-21k dataset (14 million images, 21k classes), to gain insights from different architectures and validate our framework for generalizability. To address relaxation of the equal weight constraint mentioned in section 3.1 we experiment with three ideas,

1. Relaxing the equal weight constraint but maintaining convexity (Convex). The parameters $\eta$ of the composition model $g^*$ are trained such that $\sum_i \eta_i = 1$ and $\eta_i \geq 0 \forall \eta_i \in \eta$.

2. Relaxing the convexity constraint but maintaining non negativity (Conic). The parameters $\eta$ of the composition model $g^*$ are trained such that $\eta_i \geq 0 \forall \eta_i \in \eta$.

3. Relaxing all constraints (Unconstrained).

We used the same subset of images from the ImageNet-1k to learn the composition function $g^*$ for these three variations. While the framework can be used to study any encoder layer in the model, we restrict our analysis to the last layer, whose outputs are often inputs to downstream tasks.

## 4.1 Composition Approximation: Accuracy of Learned Model on ground truth

Our initial analysis brings us back to our first question (eq. 3) posed in section 3.3. We compare the accuracy of the representations composed following (eq. 3) and that of the learned composition model. Table 1 compares the original ViT's representations, representations of the individual wavelet decompositions when summed, and the representation given by the proposed composition model. These results give us a clear picture of how the learned representations perform significantly better than just the summed representations. It can be observed that the performance for level 1 decomposition is almost on par with the original ViT model's accuracy. Also, note that the results clearly demonstrate that the learned composition function satisfies the compositionality for level 1 decomposition.

| | | | Learned | | |
|---|---|---|---|---|---|
| Model | Original | Summed | Unconstrained | Conic | Convex |
| ViT-B (Haar-level 1) | 0.792 | 0.13 | 0.775 | 0.775 | 0.771 |
| ViT-B (db4-level 1) | 0.792 | 0.13 | 0.777 | 0.775 | 0.772 |
| ViT-L (Haar-level 1) | 0.809 | 0.18 | 0.797 | 0.795 | 0.795 |
| ViT-B (Haar-level 2) | 0.83 | 0.005 | 0.51 | 0.5 | 0.48 |
| ViT-B (db4-level 2) | 0.83 | 0.005 | 0.51 | 0.51 | 0.48 |
| ViT-L (Haar-level 2) | 0.82 | 0.003 | 0.63 | 0.62 | 0.59 |

Table 1: Accuracies of original representations vs. summed representations vs. learned compositions. Note that the learned representations perform significantly better than just the summed representations.

| Model | Unconstrained | Conic | Convex |
|---|---|---|---|
| ViT-B (haar-level 1) | 0.87 | 0.87 | 0.86 |
| ViT-B (db4-level 1) | 0.9 | 0.9 | 0.89 |
| ViT-L (haar-level 1) | 0.92 | 0.91 | 0.91 |
| ViT-B (haar-level 2) | 0.53 | 0.51 | 0.49 |
| ViT-B (db4-level 2) | 0.69 | 0.68 | 0.61 |
| ViT-L (haar-level 2) | 0.65 | 0.64 | 0.61 |

Table 2: Relative accuracy of the learned composition models. Note that the target for the composed representation is the output predicted by the original image classifier (not the ground truth label).

## 4.2 Composition Approximation: Understanding the Learned Model Weights

| Model | Unconstrained | Conic | Convex |
|---|---|---|---|
| ViT-B (haar) | [ 2.02, -0.18, 0.43, 0.18] | [1.67, 0.34, 0.57, 0.02] | [0.66, 0.11, 0.10, 0.12] |
| ViT-B (db4) | [ 2.02, 0.1, -0.15, -0.16] | [1.65, 0.12, 0.63, 0.03] | [0.62, 0.09, 0.25, 0.03] |
| ViT-L (haar) | [ 1.93, 0.16, -0.02, 0.25] | [1.81, 0.28, 0.13, 0.44] | [0.68, 0.1, 0.05, 0.16] |

Table 3: Weights learned by the proposed composition model ($g^*$) for level 1 wavelet decomposition of the images.

To see how well our learned composition function $g^*$ approximates the original image's representation, we compute the relative accuracy (by considering the **original model's (ViT)** output as the ground

| Model | Unconstrained | Conic | Convex |
|---|---|---|---|
| ViT-B (haar) | [1.32, 0.35, -0.07, -0.14, 0.65, -0.20, 0.21] | [1.88, 0.61, 0.35, 0.17, 0.10, 0.10, 0.44] | [0.42, 0.13, 0.05, 0.13, 0.07, 0.10, 0.06] |
| ViT-B (db4) | [1.52, -0.18, 0.06, 0.30, 0.35, 0.16, -0.21] | [1.64, 0.40, 0.12, 0.02, 0, 0.03, 0] | [0.43, 0.11, 0.08, 0.07, 0.14, 0.06, 0.08] |
| ViT-L (haar) | [1.52, -0.01, -0.21, 0.29, 0.06, -0.01, 0.34] | [1.82, 0.29, 0.32, 0.17, 0, 0.33, 0.23] | [0.40, 0.11, 0.10, 0.08, 0.09, 0.06, 0.13] |

Table 4: Weights learned by the proposed composition model ($g^*$) for level 2 wavelet decomposition of the images.

| Model | Original Acc | Low-pass Coefficient Acc | Learned(Convex) Acc |
|---|---|---|---|
| ViT-B (haar-level 1) | 0.792 | 0.494 | 0.771 |

Table 5: Accuracy of the Original Image's representation, Low pass filtered image's representation and the learned composed representation on the testset. The results highlight the importance of other coefficients.

truth). Table 2 presents those results. It is interesting to note that the relative accuracies are similar across different constraints (variations of $g^*$). In order to investigate this further, we look at the learned model weights in Table 3 and Table 4. The learned model $g^*$ consistently weighs the approximation(Low-pass filtered image) coefficient i.e the first value, much more than the other coefficients in the representation space. Although the weights of the other coefficients are significant, it does leave some doubts whether they offer sufficient contribution. A simple experiment was conducted using the same testset to check if the approximation coefficient is enough to classify the image. The results 5 demonstrate that the other coefficients considerably improve the performance and further adds to notions of compositionality.

It is worth mentioning that there is no discernible pattern among the parameters. There is a lot of variation among the weights assigned for different $g^*$'s but the performance is quite similar. There could be multiple such compositions for an encoder layer, which leads to further questions about the representation space.

### 4.3 COMPOSITION APPROXIMATION: LEARNED RECONSTRUCTED IMAGE ANALYSIS

In this subsection, we investigate how the weights learned by the proposed composition function ($g^*$) affect the primitives (wavelet sub-bands). In other words, we apply the weights learned for the ViT encoder embeddings on their corresponding wavelet sub-bands in the image space. We consider a subset of 200 images to conduct this analysis. Table 6 presents the accuracy of the ViT model when the reconstructed images are processed. Note that although there is a significant drop in level 2 accuracies, the learned weights translate back much better for level 1 decomposition. Figure 5 visualizes the reconstructed images using Level 1 ViT-B (haar) model and Level 2 ViT-B (haar) model. The interesting thing to note is that the convex combination of the sub-bands in the image space significantly affect the pixel intensities. But the performance is still on par with other learned models.

| Model | Original | Unconstrained | Conic | Convex |
|---|---|---|---|---|
| ViT-B (haar-level 1) | 0.79 | 0.72 | 0.72 | 0.76 |
| ViT-B (db4-level 1) | 0.84 | 0.81 | 0.81 | 0.82 |
| ViT-B (haar-level 2) | 0.84 | 0.58 | 0.48 | 0.64 |
| ViT-B (db4-level 2) | 0.84 | 0.49 | 0.51 | 0.65 |
| ViT-L (haar-level 1) | 0.83 | 0.82 | 0.82 | 0.80 |
| ViT-L (haar-level 2) | 0.83 | 0.63 | 0.68 | 0.71 |

Table 6: Accuracy of ViT-B on the reconstructed images according to the weights learned by the proposed composition model ($g^*$).

### 4.4 COMPOSITION APPROXIMATION: ERROR ANALYSIS

While the classification performance of the proposed compositional model presented in the previous sections gives a broader picture, a natural question regarding the error in composition arises. In this subsection, we attempt to compare the compositional model's predictions, particularly its misclassifications, against that of the original model. Note that given the downstream task is

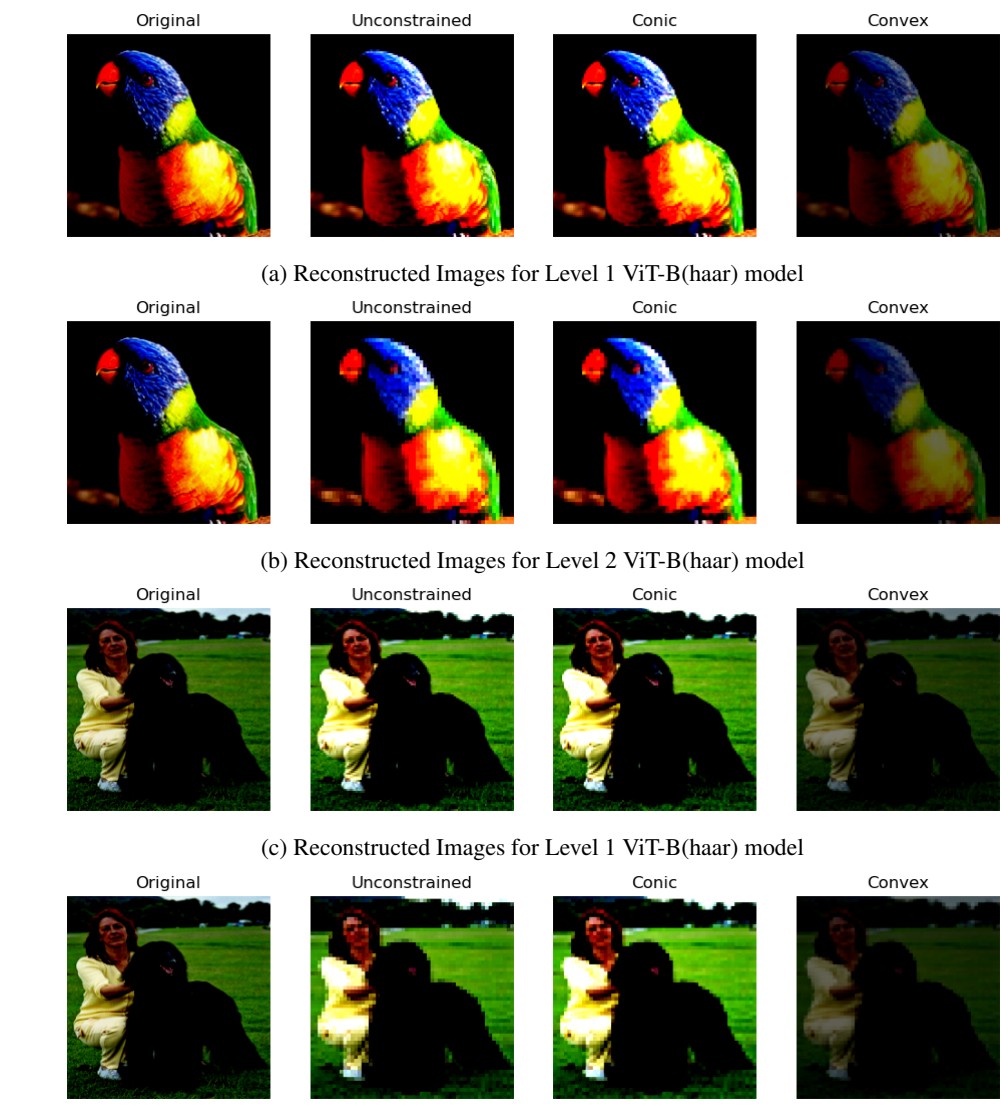

(a) Reconstructed Images for Level 1 ViT-B(haar) model

(b) Reconstructed Images for Level 2 ViT-B(haar) model

(c) Reconstructed Images for Level 1 ViT-B(haar) model

(d) Reconstructed Images for Level 2 ViT-B(haar) model

Figure 5: Reconstructed images when the weights of the learned composition model $g^*$ are applied in the input space. In other words, these reconstructions are the result of applying the weights learned by $g^*$ on the corresponding sub-bands of the original image.

classification, the error in composition is analyzed via the prediction discrepancies. We reckon it would be interesting to explore other ways to study the error in composition.

In this preliminary experiment, a sample of $1000$ images is taken from the Imagenet-1K dataset, and the performance of both the original and the compositional model (level 1 DWT decomposition) is discussed on the basis of the following.

- Percentage of images where the original model is accurate and the learned model is inaccurate ($\text{Err}_{\text{Learned}\neg\text{Org}}$).

- Percentage of images where the learned model is accurate and the original model is inaccurate ($\text{Err}_{\text{Org}\neg\text{Learned}}$).

- Percentage of images where both the models are inaccurate ($\text{Err}_{\text{both}}$).

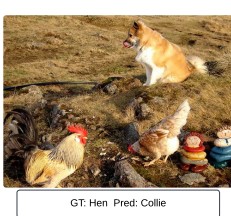 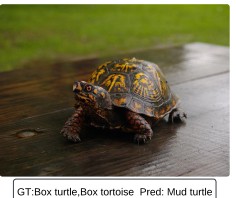

GT: Hen  Pred: Collie          GT:Box turtle,Box tortoise  Pred: Mud turtle

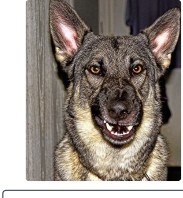 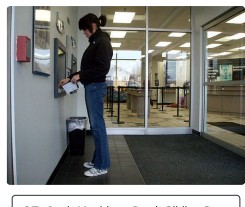

GT: German Shepherd  Pred: Elkhound          GT: Cash Machine   Pred: Sliding Door

(a) Examples images for $\text{Err}_{\text{Learned}\neg\text{Org}}$.
The erred prediction is by the learned model.

(b) Example images for $\text{Err}_{\text{Org}\neg\text{Learned}}$.
The erred prediction is by the original model.

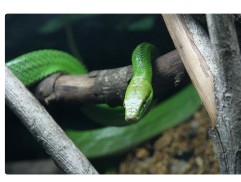 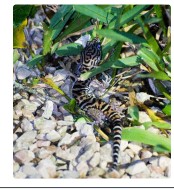

GT: Green Snake  Org Pred: Green Mamba          GT:Banded Gecko  Org Pred: Whiptail Lizard
Learned Pred: Green Mamba          Learned Pred:Alligator Lizard

(c) Sample Images for $\text{Err}_{\text{both}}$.
The erred prediction is by both models.

Figure 6: Sample Images from the test set on which the experiment was conducted.

| Model | $\text{Err}_{\text{Learned}}$ | $\text{Err}_{\text{Org}}$ | $\text{Err}_{\text{Learned}\neg\text{Org}}$ | $\text{Err}_{\text{Org}\neg\text{Learned}}$ | $\text{Err}_{\text{both}}$ |
|---|---|---|---|---|---|
| ViT-B$_{\text{Unconstrained}}$(Level-1 haar) | 19.7% | 17.1% | 3.8% | 1.2% | 15.9% |
| ViT-B$_{\text{Conic}}$(Level-1 haar) | 19.7% | 17.1% | 3.8% | 1.2% | 15.9% |
| ViT-B$_{\text{Convex}}$(Level-1 haar) | 20.4% | 17.1% | 4.3% | 1% | 16.1% |

Table 7: Errors noted on the sample test set using the learned composition model. The percentage is calculated on the basis of the sample test set (1000 images). It is important to note that there are a fraction of samples on which the learned composition performs better than the original model.

The results shown in Table 7 provide some insights regarding the learned model's performance. It is not surprising that it commits relatively more errors than the original, but it also performs better on some images. This preliminary analysis and the visual examples (Figure 6) provide sufficient motivation to further analyze the role of individual wavelet representations towards the model's prediction.

## 5 CONCLUSION AND FUTURE WORK

Our work explores notions of compositionality present in the ViT encoder layer representations. We present a general framework to measure compositional behaviour in encoder layers of ViT-based architectures. Fundamental to this framework is the use of the DWT representation as an input-dependent primitive. Our findings indicate the possibility of compositional behaviour in the ViT model. Specifically, we provide evidence for compositionality in the last encoder layer when primitives induced by a one-level DWT decomposition are applied. While our present analysis is restricted to the final encoder layer, we aim to explore all the encoder layers for potential compositionality. We hope this work leads to further analysis for explainability in ViT's.

## 6 REPRODUCIBILITY

The code for implementing the proposed compositionality framework is provided at Compositionality-in-ViT-s

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
