# OpenReview forum: "Exploring Compositionality in Vision Transformers using Wavelet Representations"
_ICLR.cc/2025/Conference — Submitted to ICLR 2025_

### Official Review · Reviewer_cTeu · 2024-10-29

**Soundness:** 3
**Presentation:** 3
**Contribution:** 2
**Rating:** 6
**Confidence:** 3

**Summary:**

The paper investigates whether representations of pretrained ViT models are compositional. The authors use DWT as the primitives (that are composed), and learn the composition (g) rather than assuming additive composition. Experiments are conducted on the final layer's output representation, and a subset of ImageNet-1k is used.

**Strengths:**

1. The paper investigates a novel approach to a significant problem. Importantly the authors show that learning a composition is significantly better than composition by addition.
2. The authors investigate the learning of composition to some extent - trying variant such as conic, convex and unconstrained.

**Weaknesses:**

1a. I believe the author's experimental setup is not using sufficient data. For reference, papers such as ViT-NeT which explores interpretability of ViT uses three datasets each of which is 10-20k images total. I'd suggest authors scale up their datasets - if we choose to use ImageNet-1k, a test set of at least 10 images per class would be more convincing. Currently, with a 15% test fraction, each class gets 1-2 images. I hesitate to make conclusions based on such a small test set per class.

1b. No significant analysis is done on the test dataset, nor of the errors that composition leads to. It would be beneficial to provide some examples where composition lead to large error and some examples where the error is minimal. For example, Haar DWT is great when we have sudden transitions in signal - like sharp images. Perhaps blurrier examples or classes would cause larger error?

2. Authors do not inspect intermediate layers of ViT, which is a glaring hole to me - while it is true that most downstream tasks will use the penultimate layer of the ViT, I am still left wondering if all intermediate representations are composable, if its just some of them, is compositionality lost at some levels of the stack? I would recommend repeating the same analyses using intermediate layers’ representations.

**Questions:**

1. Could the authors clarify how compositionality with discrete wavelet transforms leads to better explainability? Some examples or related work would help. I can see some intuitive argumentation but I think it has not been articulated in the paper.

2. The observation that convex combination of sub-bands in image space alters pixel values significantly is interesting. Why does it not degrade accuracy? Is it because of normalization?

---

> ### Author Response · Authors · 2024-11-28
> **Official comment to Reviewer cTeu: Part One**
>
> We thank the reviewer for their assessment and we hope to answer their questions point by point.
>
> ### **Weaknesses**
>
> > **1.a)**  I believe the author's experimental setup is not using sufficient data. For reference, papers such as ViT-NeT which explores interpretability of ViT uses three datasets each of which is 10-20k images total. I'd suggest authors scale up their datasets - if we choose to use ImageNet-1k, a test set of at least 10 images per class would be more convincing. Currently, with a 15% test fraction, each class gets 1-2 images. I hesitate to make conclusions based on such a small test set per class.
>
> We thank the reviewer for this comment. Please note that the parameters we are training for the compositional model are merely a handful. Also, we believe ImageNet is one of the most complex datasets that offers a lot of variety among the samples. In the revision, we scaled up the dataset to 50000 samples (1000 classes,50 images/class) with a 60:20:20 train:val:test split. So the test set contains 10000 samples (each class gets 10 images) and repeated some of the experiments. Due to time constraints, we could only carry out the experiments for level 1 decomposition. We plan to carry out the rest of the level 2 experiments with the scaled-up dataset. The results for Level 1 have been updated in the revised draft.
>
> **Comparing the accuracy of different composition models $g^{*}$ on the testset:**
>
> |       Model        |  Original   |  Average  |  Unconstrained  |  Conic   |  Convex  |
> | ------------------- | ------------- | ------------ | --------------------- | ---------- | ------------ |
> | ViT-B(haar level-1) |  0.792 | 0.13 | 0.775| 0.775 | 0.771 |
> |ViT-L(haar level - 1) | 0.809 | 0.18 | 0.797 | 0.795 | 0.795|
> |ViT-B (db4 level - 1) | 0.792 | 0.13 | 0.777 | 0.775 | 0.772 |
>
> **Relative accuracy of the learned composition models. Note that the target for the composed representation is the output predicted by the original image classifier (note the ground truth label).**
>
> |       Model        |  Unconstrained  |  Conic   |  Convex  |
> | ------------------- | --------------------- | ---------- | ------------ |
> | ViT-B(haar - level 1) |  0.875 | 0.873 | 0.862 |
> | ViT-B( db4 -level 1) | 0.90 | 0.904 | 0.898 |
> | ViT-B( haar - level 1) | 0.918 | 0.916 | 0.911|
>
> > **1.b)** No significant analysis is done on the test dataset, nor of the errors that composition leads to. It would be beneficial to provide some examples where composition lead to large error and some examples where the error is minimal. For example, Haar DWT is great when we have sudden transitions in signal - like sharp images. Perhaps blurrier examples or classes would cause larger error?
>
> We appreciate the reviewer's point. We have conducted a preliminary experiment (Section 4.4 in the revised draft) to identify the samples on which the learned model fails. However, a deeper analysis to test the robustness of compositionality is currently out of scope of this paper.  We would like to emphasize that the primary goal of this work is to present a framework to check if notions of compositionality are present in the representation space and the level 1 decomposition results provide strong support.
>
> > **2)**  Authors do not inspect intermediate layers of ViT, which is a glaring hole to me - while it is true that most downstream tasks will use the penultimate layer of the ViT, I am still left wondering if all intermediate representations are composable, if its just some of them, is compositionality lost at some levels of the stack? I would recommend repeating the same analyses using intermediate layers’ representations.
>
> The intermediate layer representation of each image is of size 197x768. The cls token is of size 1x768. Since the input to the classification layer is just the cls token, performing this analysis on the final encoder layer is easier. There is no trivial way to classify the intermediate layer's output. To perform the analysis on intermediate layers would require storing the entire 197x768 representation for each image and its subsequent wavelet components in order to train the model. Due to memory and computational limitations we could not perform the analysis for the intermediate representations.

---

> ### Author Response · Authors · 2024-11-28
> **Official comment to Reviewer cTeu : Part Two**
>
> ### **Questions**
>
> > **1)** Could the authors clarify how compositionality with discrete wavelet transforms leads to better explainability? Some examples or related work would help. I can see some intuitive argumentation but I think it has not been articulated in the paper.
>
> Given the challenging nature of composing 'semantic' primitives for object recognition, the paper presents DWT as a potential framework for analyzing compositionality in ViTs.
> * The analysis presented in the paper explains the relative importance given to the individual frequency bands by the ViT models. This may lead to understanding the object recognition task from the frequency domain perspective.
> * The presented compositionality framework enables the incorporation of any specific domain knowledge with respect to the primitives (in this case, the DWT sub-bands) towards the downstream task. In other words, one can train models that can give specific weights to the sub-bands while learning to solve the task if needed.
> * In the future, the community may bridge the frequency-to-semantics gap with the availability of sophisticated tools so that the presented framework can directly compose to the semantics
>
>
> > **2)** The observation that convex combination of sub-bands in image space alters pixel values significantly is interesting. Why does it not degrade accuracy? Is it because of normalization?
>
> This is definitely interesting. Probing the combinations themselves revealed that the $l2$ norm between the convex combination and the original image is lower than the other two combinations. We hesitate to say it is because of normalization and this would require further analysis to answer.

---

### Official Review · Reviewer_iA5z · 2024-11-04

**Soundness:** 2
**Presentation:** 2
**Contribution:** 2
**Rating:** 5
**Confidence:** 3

**Summary:**

The paper examines a type of compositionality in the representations of vision transformers (ViT). Building on the compositionality concept proposed by Andreas (2019), the authors apply it to Discrete Wavelet Transform (DWT) representations. Empirical results suggest that the last layer of a transformer displays a certain degree of compositionality for a one-level DWT of the input.

**Strengths:**

* The paper is generally well-written.
* It builds on the well-established framework by Andreas (2019) for compositional representations.
* Wavelets are a common and natural basis for image representations in signal processing.

**Weaknesses:**

* My main concern is that the presented results do not convincingly demonstrate compositionality. Rather than defining true combinations of wavelet primitive representations, it appears that the learned weights mainly select the low-pass filtered image (Table 3). Indeed, it is not particularly surprising that the images in Figure 5 perform similarly to the original images.
* Compositionality is typically more valuable when components are semantic rather than appearance-based. It is doubtful that wavelets would exhibit compositional properties in the final layers of a model, where higher-level concepts are typically captured; instead, this is more likely to occur in lower-level layers. Furthermore, the idea that wavelets are a good basis for compositional representations is not really explored, and no other decomposition methods are considered or compared.

Minor:
* The organization of the paper could be improved, since details on the framework (Section 3.3) are presented after the initial discussion on quantifying compositionality (Section 3.1). Additional background on wavelet decompositions to clarify notation (e.g., LL, LH, etc., in Figure 4) would also be helpful.
* While it can be guessed, the exact meaning 'convex' and 'conic' relaxations is never explained. Other details could be clarified (see below)

**Questions:**

* Could the authors clarify why they believe their results demonstrate compositionality, why the DWT is considered a suitable feature decomposition, and why compositional behavior would be expected in the last layer?
* l.299: "The target is the final classification layer output of the original image" does this mean using an L2 loss?
* l.445: Why is the analysis conducted across different settings?

Minor:
* l.237 I believe $E_L$ should be $E_l$.

---

> ### Author Response · Authors · 2024-11-27
> **Official Response to Reviewer iA5z : Part One**
>
> We thank the reviewer for their constructive feedback. We hope to answer their queries point by point.
>
> ### **Weaknesses**
>
> > **1)**  My main concern is that the presented results do not convincingly demonstrate compositionality. Rather than defining true combinations of wavelet primitive representations, it appears that the learned weights mainly select the low-pass filtered image (Table 3). Indeed, it is not particularly surprising that the images in Figure 5 perform similarly to the original images.
>
> We thank the reviewer for this comment. As mentioned in section 3.1 of the draft, our motivation for a learned composition of the
> wavelet primitives instead of the true combination (summing each primitive) of primitives in the representation space is based on our
> initial experiments of SSIM scores and CKA plots. These results show that the “true” combination does not demonstrate compositional
> behavior. We then speculate that if any compositional behavior does exist, could it be learned? Our results confirm that such a composition exists for the final encoder layer for level 1 DWT decomposition. To answer the reviewer’s concern as to the selection of mainly the low-pass filtered image, we conducted an experiment to see how the low-pass filtered image’s representation performs compared to the original image’s representation. We take 10 images per class from all the 1000 classes in ImageNet-1K and present the results.
>
> |  Model                |   DWT Level  | Original Accuracy  |  Low-pass filtered Image Accuracy  |  Learned Accuracy |
> | ---------------------- |   --------------- | ------------------------  | ---------------------------------------------- | ------------------------- |
> | ViT Base (Unconstrained)            |    Level 1      |         0.792              |                       0.494                         |           0.771            |
>
> Although the learned composition pays more attention to the low-filtered component, the other components are also important. This clearly signifies that a composition of all these components does indeed approximate the original representation much better than the low-pass component.
>
> > **2)**  Compositionality is typically more valuable when components are semantic rather than appearance-based. It is doubtful that wavelets would exhibit compositional properties in the final layers of a model, where higher-level concepts are typically captured; instead, this is more likely to occur in lower-level layers. Furthermore, the idea that wavelets are a good basis for compositional representations is not really explored, and no other decomposition methods are considered or compared.
>
> We understand that probing compositionality is done by identifying simpler concepts that can be composed into complex ideas. In the
> NLP setting, it is easy to find this analogy, which is that words can form simpler concepts composed into complex sentences. However,
> it is difficult to break down in the context of vision when the complex ideas are images. The primitive set required for the framework is very
> challenging to identify since the image space is continuous. We chose the DWT to extract the primitives because of the sound mathematical proof that wavelet decompositions have perfect reconstruction. We check for a homomorphism between the input and representation space on the foundation that the DWT has perfect reconstruction in the input space. To the best of our knowledge, we are the first to probe for compositionality with the framework presented in our paper using wavelets. Other decomposition methods, such as Fourier transform or Discrete Cosine Transform (DCT), also exhibit lossless decomposition and reconstruction, but they do not preserve spatial information, i.e., the frequency spectrum of an image is not visually meaningful. Hence, we chose DWT as the method for decomposition for our framework.

---

> ### Author Response · Authors · 2024-11-27
> **Official Response to Reviewer iA5z : Part two**
>
> ### **Questions**
>
> > **1)**  Could the authors clarify why they believe their results demonstrate compositionality, why the DWT is considered a suitable feature decomposition, and why compositional behavior would be expected in the last layer?
>
> The framework presented in the paper divides the input image into its DWT primitives. The original image and its primitives are passed
> through the model separately. The individual primitives are linearly combined using the weights of the learned compositional model and is
> then passed through the final classification layer. The results demonstrate that composed representations perform similarly to the original image representation. We chose the DWT to extract the primitives because of the sound mathematical foundation that wavelet decompositions have perfect reconstruction. On the basis that a perfect composition exists in the input space, we investigate if a homomorphism exists between the input and representation space learned by the ViTs. While our analysis demonstrates the notions of compositionally (as per the framework introduced by Andreas 2019) with respect to the DWT primitives, more needs to be understood
> about why it is manifested.
>
> > **2)** l.299: "The target is the final classification layer output of the original image" does this mean using an L2 loss?
>
> We use the Cross Entropy loss between the classification layer output of the original image’s representation (i.e., predicted soft labels)
> and the classification layer output of the composed representation.
>
> > **3)** l.445: Why is the analysis conducted across different settings?
>
> We regret this confusion. The analysis was conducted using the same test set. At the time of submission, we could only run a part of the
> test set for some of the results. We have now revised the draft such that the updated results follow the same setting.

---

### Official Review · Reviewer_guFm · 2024-11-05

**Soundness:** 3
**Presentation:** 3
**Contribution:** 2
**Rating:** 5
**Confidence:** 3

**Summary:**

This paper presents experiments exploring the use of the Discrete Wavelet Transform (DWT) for evaluating compositionality within ViTs.

The work is motivated by prior work (Andreas 2019) which proposed a general framework (Tree Reconstruction Error, TRE) for measuring compositionality as a homomorphism between the input space and a representation space.

Because images do not have an obvious set of primitives in the same way that tokenized word vocabularies do in language spaces, this paper proposes to use DWT components as the primitive representation.

In the first set of experiments, the paper argues that the naive addition of the DWT components will not necessarily yield a compositional representation, and instead proposes to learn a composition function, represented as a weighted sum of the components.

Experiments applying DWT to the final layer of a ViT are also presented, showing that approximating the ViT representation as the weighted sum (under the learned composition function) of its inverse DWT components yields comparable performance to the model’s output under the original representation.

I have concerns about the soundness of the experimental setup (see Weaknesses). Given the relaxation introduced in L214-215, it’s not clear to me that the provided experimental results can actually soundly support claims about compositionality since the homomorphism is no longer defined in relation to the input space.

**Original Recommendation**:
I think the paper presents a compelling idea, but the paper could be greatly strengthened by improvements to clarity and substantiation of claims. I would value hearing the authors’ response to my concerns before finalizing my score, but in its current form I am recommending that the paper be rejected.

**Revised Recommendation (11/29)**:
The authors have largely addressed my concerns, and so I have revised my overall score to 5. To conclude my score, I agree with some of the concerns that iA5z raised related to how compositionality is generally understood in the community through the lens of semantics rather than appearance, which I don't believe were completely sufficiently addressed. I think the impact of the paper could be stronger with some following changes which I believe would likely constitute a new submission rather than minor revisions:

* (1) The framing of the paper could modified to better motivate why studying the type of compositionality presented is useful. Something that comes to mind is that text-based natural language is *already* an abstracted signal (and which is tied to semantics), perhaps it would make sense to contextualize this paper within compositionality literature for audio signals rather than text.
* (2) The introduction of further experiments showing how this type of compositionality can be useful in downstream tasks would greatly help to further motivate this approach. For semantic based tasks, compositional representations tend to yield improvements in generalization capabilities of models -- can something analogous to this be shown for DWT based compositional representations?

In other words, I think the motivation/contribution of the paper would be strongest if it could be shown either that the DWT representations are tied to semantic compositionality, or if not, that this kind of compositionality is still explicitly useful downstream.

**Strengths:**

* The paper presents a compelling idea — that DWT components could serve as primitives through which to study compositionality in ViTs.

**Weaknesses:**

Weaknesses:

**Clarity**: The paper can be unclear at times over specifics of what was done or how, I think that further elaboration from the authors throughout could help strengthen the paper.
* Figure 1: What does it mean here for the original image’s representation to be compared to the composed image representation? Does it mean that the maps we see in the figure are the result of of the comparison, or are these just the composed representations and the comparison was done outside of the figure?
* Section 3.1 ViT: What was the ViT model used for this experiment? What was it pre-trained on? The section does not explicitly specify these experimental conditions.
* L214-L215: What is the purpose of f_eta ? I understand that this is defined in Andreas 2019, but its definition is missing from this paper. I think explicitly defining it and its role would improve the clarity of this section.
* Figure 3: What do the C’s and a’s represent?
* Figure 4: What are LL, LH, HL, and HH?
* In general, I think the clarity of the paper could be improved by an additional grammar/phrasing pass.

**Soundness**: I have a few concerns regarding the soundness of the experimental setup, and I am unsure if the paper’s claims are supported. I would value hearing the authors' response to the following points/questions:
* Section 3.1: I believe this section aims to show that simply adding the DWT wavelets with equal weights does not yield a correct composition. However, this claim is supported by showing results over a single image from a single ViT model. I believe this experiment would need a much larger sample size over images/models in order to make such a broad claim.
* L216-217: The original compositionality formulation from Andreas 2019 is modified to shift the application of the encoder function from the input space to an arbitrary intermediate representation space within the model. If I’m understanding correctly, doesn’t this violate the core premise of the problem statement? The purpose of compositionallity tests is to find homomorphisms between the **input space** and the representation space, this is important because the input space comes from the data generating function. I’m not sure if it makes sense for this test to be defined for the transformation from one hidden layer to another. Minor, but I would suggest modifying the sentence “instead of drawing exact parallels, we tweak this statement to suit our analysis”, since the wording gives the impression that the formulation was modified to suit the narrative of the paper, rather than the needs of the original empirical question being asked.

**Conclusions/Takeaways**: I believe that the provided experimental results show that the representations of the final ViT layer can be represented as a linear combination of its inverse DWT components. However, given my concerns stated above with soundness, it’s not clear to me what this entails about compositionality.

**Questions:**

Please see questions under Weaknesses.

---

> ### Author Response · Authors · 2024-11-27
>
> We thank the reviewer for their detailed response and valuable feedback. We hope to answer all of their points in similar fashion.
>
> ## **Weaknesses**
> ### **Clarity**
> > **1)** Figure 1: What does it mean here for the original image’s representation to be compared to the composed image representation?
> Does it mean that the maps we see in the figure are the result of the comparison, or are these just the composed representations and the
> comparison was done outside of the figure?
>
> The SSIM maps shown in the figure are the results **after** comparison of the original representation and the composed representation at every layer of the encoder. We have included the clarification in the caption of the figure in the revised draft.
>
> > **2)** Section 3.1 ViT: What was the ViT model used for this experiment? What was it pre-trained on? The section does not explicitly specify
> these experimental conditions.
>
> We thank the reviewer for pointing this out, and we have included the clarifications (L321-L322) in the revised draft. To answer this query, the models used are ViT- Base and ViT Large, both of which are pre-trained on the Imagenet-21k dataset.
>
> > **3)** L214-L215: What is the purpose of f_eta ? I understand that this is defined in Andreas 2019, but its definition is missing from this paper.
> I think explicitly defining it and its role would improve the clarity of this section.
>
> We thank the reviewer for pointing this out, and we have included the clarifications in the revised draft. The clarifications are included at L200-L201 and L225-L226 in the revised draft. To answer the question $\hat{f_{\eta}}$ is a compositional approximation of the complex model $f$ . It can be used to measure the compositionality of the model.
>
> > **4)** What do the C’s and a’s represent?
>
> The Ca's and Cd's represent the approximate and detail coefficients for each level of decomposition. The approximate is the output
> of a low pass filter and the detail is the output of a high pass filter. For the next level, the approximate is passed through the low pass and high pass filter to get the coefficients for the next level.
>
> > **5)** What are LL, LH, HL, and HH?
>
> LL,LH,HL and HH are the corresponding Low-Low, Low-High, High-Low and High-High subbands from the wavelet decomposition. We have edited the figure and corrected the labeling. The labels for each image now follow the previously defined notations.
>
> ### **Soundness**
>
> > **1)**  Section 3.1: I believe this section aims to show that simply adding the DWT wavelets with equal weights does not yield a correct composition. However, this claim is supported by showing results over a single image from a single ViT model. I believe this experiment
> would need a much larger sample size over images/models in order to make such a broad claim.
>
> This is a minor confusion. The SSIM map (Figure 1) was computed over a representative image but plot (in the initial submission) showing the CKA scores (Figure 2) takes 200 images and averages their performance over all encoder layers. Taking the reviewer’s point into account we have conducted another experiment by taking a total of 10000 images (10 images/per class from imagenet-1k dataset). The current figure in the revised draft shows these results.
>
> > **2)** L216-217: The original compositionality formulation from Andreas 2019 is modified to shift the application of the encoder function
> from the input space to an arbitrary intermediate representation space within the model. If I’m understanding correctly, doesn’t this
> violate the core premise of the problem statement? The purpose of compositionality tests is to find homomorphisms between the input
> space and the representation space, this is important because the input space comes from the data generating function. I’m not sure if it
> makes sense for this test to be defined for the transformation from one hidden layer to another. Minor, but I would suggest modifying the
> sentence “instead of drawing exact parallels, we tweak this statement to suit our analysis”, since the wording gives the impression that the
> formulation was modified to suit the narrative of the paper, rather than the needs of the original empirical question being asked.
>
> We thank the reviewer for this observation. We regret the confusion caused by overlooking it. The proposed framework investigates the homomorphism from the input space to the embedding space learned by the ViTs. We have rectified the mistake in the
> revised draft (L228-L234). To clarify, the model is indeed taking inputs from the input space (which is the image space), whose derivatives (wavelet decompositions) are fed to the model $f$ . To check for the compositionality of layer $l$, the encoder representations of that layer (transformation from the input to that layer) are fed to the composition function $\hat{f_{\eta}}$

---

> > ### Comment · Reviewer_guFm · 2024-11-29
> >
> > Thank you to the authors for addressing my concerns and for clearing up confusion. I have revised my overall score to 5 (see update in my review for more details).

---

### Author Response · Authors · 2024-11-28
**Major Changes made in the revised draft**

We thank all the reviewers for their constructive feedback. Based on their comments we have made some changes in the revised draft. They are listed as follows:

* The  experiments have been repeated with a significantly larger dataset (50k images from ImageNet-1k dataset with a train:val:test split of 60:20:20, previous results were for 10k images) for reliable results. Due to the limited time, we could only repeat the
 Level 1 experiments and we aim to complete the Level 2 results soon. **Only the level 1 results have been updated** in the revised draft.
* The analogy regarding the compositional approximation has been improved (L228-L234)
* LL ,LH , HL ,HH notations have been defined (L153 - L154)
* A new experiment has been added (Section 4.4) to analyze the errors of the composition approximation.

---

### Meta-Review · Area_Chair_PaJ1 · 2024-12-17

**Metareview:**

(a) Summary

This paper investigates how to test compositionality in ViT encoder. It presents a framework with compositionality setting initially proposed by Andreas 2019, and employs the Discrete Wavelet Transform (DWT) for analyzing the compositional structure of the vision input. The experiments indicate that the primitives with a one-level DWT decomposition  as input leads to a certain degree of compositionality in the last layer of ViT encoder.

(b) Strengths
+ It is well motivated: the paper investigates an important problem in learning compositional representation.
+ It is based on well-established work by Andreas.
+ Wavelets are a common & natural way to encode image into primitives.

(c) Weaknesses
- The experimental setup is weak: this experiments need a much larger sample size over images/models in order to make such a broad claim on compositionality.
- The experimental results do not support the claims: the presented results do not convincingly demonstrate compositionality.
- Wavelets are for low-level representation not for higher-level layers compositionality.
- It does not compare with other decompostion methods.
- The clarity of the paper could be improved.

(d) Decision

The investigation in this paper for understanding compositionality in ViT encoder layers is well-motivated. However, the experimental setting and results are not strong enoughful to support the claims on compositionality of encoders. I think the paper is not ready for publication in its current form due to the weaknesses listed in the summary.
Please keep the reviewer comments in mind when preparing a future version of the manuscript.

**Additional Comments On Reviewer Discussion:**

The reviewers agreed that the paper's proposal for employing DWT as primitives for studying compositionality in ViT is novel and interesting. They also shared the same concerns on the weak experimental setup, the upsupported claims from the experimental results, and the clarity of the paper. Although the authors' rebuttal and updated manuscript addressed some concerns, the reviewers still think the paper needs another round of revision and review to make it stronger.

---

### Decision · Program_Chairs · 2025-01-22

Reject